# Learning Object-Centric Representations of Multi-Object Scenes from Multiple Views

**Li Nanbo**
School of Informatics
University of Edinburgh
nanbo.li@ed.ac.uk

**Cian Eastwood**
School of Informatics
University of Edinburgh
c.eastwood@ed.ac.uk

**Robert B Fisher**
School of Informatics
University of Edinburgh
rbf@inf.ed.ac.uk

## Abstract

Learning object-centric representations of multi-object scenes is a promising approach towards machine intelligence, facilitating high-level reasoning and control from visual sensory data. However, current approaches for *unsupervised object-centric scene representation* are incapable of aggregating information from multiple observations of a scene. As a result, these "single-view" methods form their representations of a 3D scene based only on a single 2D observation (view). Naturally, this leads to several inaccuracies, with these methods falling victim to single-view spatial ambiguities. To address this, we propose *The Multi-View and Multi-Object Network (MulMON)*[1]—a method for learning accurate, object-centric representations of multi-object scenes by leveraging multiple views. In order to sidestep the main technical difficulty of the *multi-object-multi-view* scenario—maintaining object correspondences across views—MulMON iteratively updates the latent object representations for a scene over multiple views. To ensure that these iterative updates do indeed aggregate spatial information to form a complete 3D scene understanding, MulMON is asked to predict the appearance of the scene from novel viewpoints during training. Through experiments we show that MulMON better-resolves spatial ambiguities than single-view methods—learning more accurate and disentangled object representations—and also achieves new functionality in predicting object segmentations for novel viewpoints.

## 1 Introduction

Traditional VAEs[15] use "single-object" or "flat" vector representations that fail to capture the compositional structure of natural scenes, i.e. the existence of interchangeable objects with common properties. As a result, "multi-object" or object-centric representations have emerged as a promising approach to scene understanding, improving sample-efficiency and generalization for many downstream applications like relational reasoning and control [5, 20, 3]. However, recent progress in unsupervised object-centric scene representation has been limited to "single-view" methods which form their representations of 3D scenes based only on a single 2D observation (view). As a result, these methods form inaccurate representations that fall victim to single-view spatial ambiguities (e.g. occluded or partially occluded objects) and fail to capture 3D spatial structures.

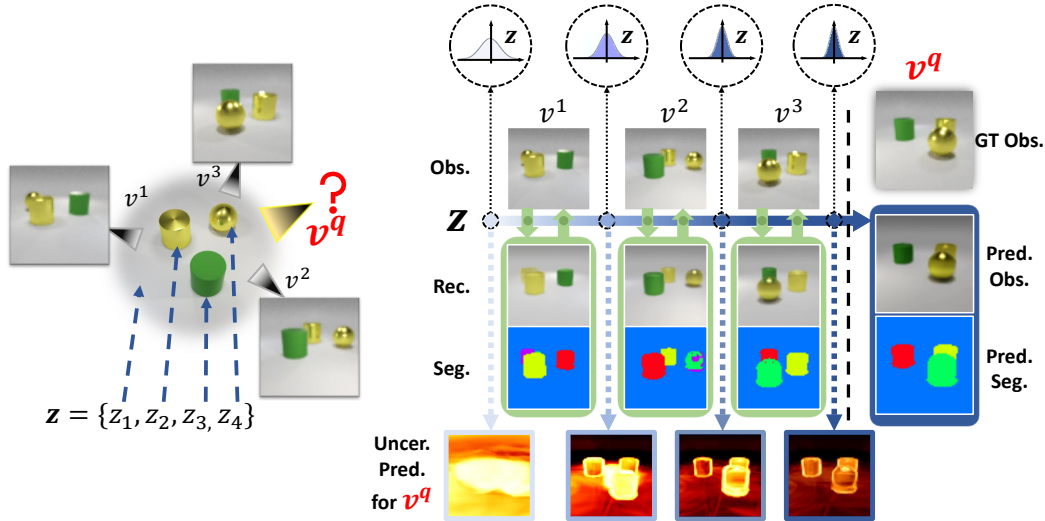

Figure 1: **Left:** *Multi-object-multi-view setup.* $v^q$ denotes the query viewpoint, while $z_k$ denotes "slot" $k$, i.e. the latent object representation of a scene object. **Right:** *MulMON overview.* Starting with a standard normal prior, MulMON iteratively refines $\mathbf{z}$ over multiple views, each time reducing its uncertainty about the scene——as illustrated by the darkening, white-to-blue arrow. Within-view "inner loop" iterations are depicted by the green arrows and boxes. Cross-view "outer-loop" iterations are depicted by the white-to-blue arrows and boxes. At the bottom, we have visualised MulMON's reduction in uncertainty about $\mathbf{z}$ in image space, where each image shows the per-pixel variance of MulMON's predicted observation from query viewpoint $v^q$. MulMON's final predictions for $v^q$ (observation and segmentation) are shown to the right of the vertical dotted line.

To address this, we present MulMON (Multi-View and Multi-Object Network)—an unsupervised method for learning object-centric scene representations from multiple views. Using a spatial mixture model [10] and iterative amortized inference [19], MulMON sidesteps the main technical difficulty of the multi-object-multi-view scenario—maintaining object correspondence across views—by iteratively updating the latent object representations for a scene over multiple views, each time using the previous iterations posterior as the new prior. To ensure that these iterative updates do indeed aggregate spatial information, rather than simply overwrite, MulMON is asked to predict the appearance of the scene from novel viewpoints during training. Given images of a *static* scene from several viewpoints, MulMON forms an object-centric representation, then uses this representation to predict the appearance and object segmentations of that scene from unobserved viewpoints. Through experiments we demonstrate that:

- MulMON better-resolves spatial ambiguities than single-view methods like IODINE [9], while providing all the benefits of object-based representations that "single-object" methods like GQN [8] lack, e.g. object segmentations and manipulations (see Section 5).

- MulMON accurately captures 3D scene information (rotation along the vertical axis) by integrating spatial information from multiple views (see Section 5.3).

- MulMON achieves both inter- and intra-object disentanglement—enabling both single-object and single-object-property scene manipulations (see Section 5.3).

- MulMON represents the first feasible solution to the *multi-object-multi-view problem*, permitting new functionality like viewpoint-queried object-segmentation (see Section 5.2).

## 2 Background

### 2.1 Multi-object representations

Assuming a single observation or view $x^1$ for now, we can formally describe the goal of "multi-object" (i.e. object-centric) scene learning as that of computing the posterior $p(z_1, z_2, ..., z_K|x^1)$, where

$K$ is the number of 3D objects in the scene including the background "object", $z_k \in \mathbb{R}^D$ is a $D$-dimensional latent representation of object $k$, and $x^1 \in \mathbb{R}^M$ is a 2D image with $M$ pixels. As $K$ is unknown, recent "multi-object" works [4, 9] have fixed $K$ globally to be a number that is sufficiently large (greater than actual number of objects) to capture all the objects in a scene and allowing for empty slots. Thus, we will use $K$ to represent the number of object *slots* hereafter.

## 2.2 Multi-object representations from multiple views

As 2D views of 3D scenes are inherently under-specified, they often contain spatial ambiguities (e.g. occlusions or partial occlusions). As a result, "single-view" methods often learn inaccurate representations of the underlying 3D scene. To solve this, it is desirable to aggregate information from multiple views into a single accurate representation of the (static) 3D scene. Recently, this was achieved with some success by GQN for single-object (i.e. single-slot) representations. However, doing so for multi-object representations is much more challenging due to the difficultly of the object matching problem, i.e. maintaining object-representation correspondences across views. As a result, this *multi-object-multi-view* (MOMV) problem remains unsolved.

With this in mind, we can define a more general object-centric scene representation learning problem as that of learning a representation of a $K$-object scene based on $T$ uncalibrated observations from random viewpoints, where the scenes are static and assumed to be a spatial configuration that is independent of the observer. Formally, this involves computing the posterior $p(z_1, z_2, ..., z_K | x^1, x^2, ..., x^T)$. Since a 2D observation of a 3D scene must be associated with a viewpoint, we can better specify the problem as that of computing $p(z_1, z_2, ..., z_K | x^1, x^2, ..., x^T, v^1, v^2, ..., v^T)$, where $v^t \in \mathbb{R}^J$ is the viewpoint associated with image $x^t$. This can be written more compactly as $p(\mathbf{z} = \{z_k\} | \{(x^t, v^t)\})$, where both the latent object set $\mathbf{z} = \{z_k\}$ and observation set $\{(x^t, v^t)\}$ are permutation-invariant.

# 3 Method

Our goal is to learn structured, object-centric scene representations that accurately capture the spatial structure of 3D multi-object scenes, and to do this by leveraging multiple 2D views. Key to achieving this is 1) an outer loop that iterates over views, aggregating information while avoiding the object matching problem, and 2) a training procedure that ensures that these outer loops are indeed used to form a complete 3D understanding of the scene, rather than just overwriting each other. We detail 1) in Section 3.1, and 2) in Section 3.4. Additionally, we describe the viewpoint-conditioned generative model and iterative inference procedure in Sections 3.2 and 3.3 respectively.

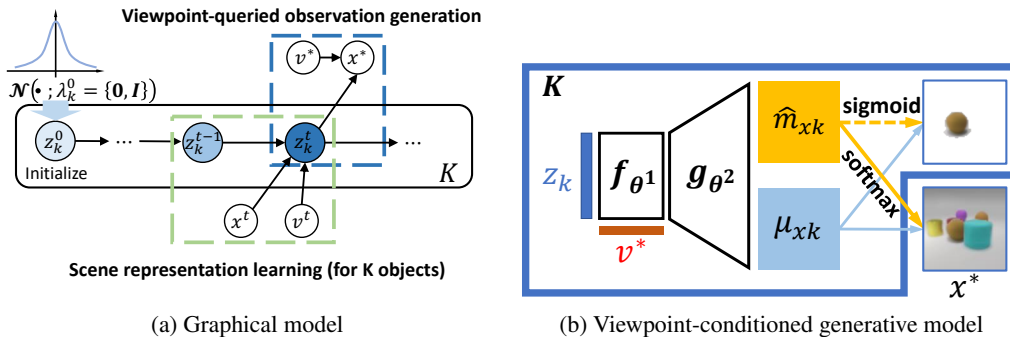

(a) Graphical model        (b) Viewpoint-conditioned generative model

Figure 2: (a) Graphical view of MulMON's cross-view iterations. The two core components, a viewpoint-conditioned generative model (Section 3.2) and an inference model (Section 3.3), are shown in blue and green boxes respectively. (b) For viewpoint-conditioned generation, each of the $K$ latent object representations $\mathbf{z}_k$ are transformed w.r.t. a viewpoint $v^*$ using the function $f_{\theta^1}$, before being passed through a decoder $g_{\theta^2}$ to render a viewpoint-queried observation $x^*$. As shown, $g_{\theta^2}$ actually outputs the pixel-wise means $\mu_{xk}$ and predicted object masks $\mathbf{softmax}(\hat{m}_{xk})$ of the spatial Gaussian mixture model, which are combined to get $x^*$.

### 3.1 Iterating over views

**Cross-view iterations (outer loop).** For a static scene, we consider that the latent scene representation $\mathbf{z} = \{z_k\}$ is updated sequentially in $T$ steps as the $T$ observations are obtained one-by-one from $t = 1$ to $t = T$, where $t$ denotes the updating step. This suggests that $\mathbf{z}^t$ is obtained by updating $\mathbf{z}^{t-1}$ using a new observation $x^t$, taken at viewpoint $v^t$ (see the green box in Figure 2a). Therefore, by making an assumption that $\mathbf{z} = \mathbf{z}^t$ for any integer $t \in [1, T]$, we can compute the target multi-view posterior in a recursive form as:

$$p(\mathbf{z} = \{z_k\}|x^{1:T}, v^{1:T}) = p(\mathbf{z}^0) \prod_{t=1}^{T} p(\mathbf{z}^t|x^t, v^t, \mathbf{z}^{t-1}), \tag{1}$$

where $z_k^{t-1}$ is the latent representation of a scene object (indexed by $k$) *before* observing $x^t$ at viewpoint $v^t$, $z_k^t$ the representation afterwards. and $p(\mathbf{z}^0)$ the initial guess which we assume to be a standard Gaussian distribution $\mathcal{N}(\mathbf{0}, \mathbf{I})$. The formulation in equation 1 turns the multi-view problem into a recursive single-view problem and, in theory, enables online learning of scenes from an infinitely large number of observations without causing memory overflow.

**Within-view iterations (inner loop).** As shown in Figure 2, MulMON consists of a scene-representation inference model and a viewpoint-conditioned generative model. In each iteration, the inference model starts with a prior assumption about the $K$ objects in the latent space, i.e. $\mathbf{z} = \mathbf{z}^{t-1} = \{z_k^{t-1}\}$, and approximates the target posterior $p(\mathbf{z}^t = \{z_k^t\}|x^t, v^t, \mathbf{z}^{t-1})$ after observing $x^t$ at viewpoint $v^t$. The approximation, as mentioned in Section 1, is handled by iterative amortized inference[19] and the approximate posterior is passed to the next iteration as the prior assumption. Therefore, a single iteration is a single-view process that takes a latent prior about $\mathbf{z}^{t-1} = \{z_k^{t-1}\}$ and an image observation $x^t$ (taken at a viewpoint $v^t$) as inputs. We call the single-view iterative process the *inner loop*, and the cross-view Bayesian updating process (see equation 1) the *outer loop*.

### 3.2 Generative Model

We model an image $x^t$ with a spatial Gaussian mixture model[23, 10], similar to MONet[4] and IODINE[9], and additionally we take as input (condition on) the viewpoint $v^t$. We can then write the generative likelihood as:

$$p_\theta(x^t|\mathbf{z}^t, v^t) = \prod_{i=1}^{M} \sum_{k=1}^{K} p_\theta(C_i^t = k|z_k^t, v^t) \cdot p_\theta(x_{ik}^t|z_k^t, v^t), \tag{2}$$

where $x_{ik}^t$ are the RGB values in image $t$ at at a pixel location $i$ that pertain to object $k$, $p_\theta(x_{ik}^t|z_k^t, v^t)$ is the Gaussian density function parametrized by a neural network $\theta$, and $m_{ik}$ is the mixing coefficient for object $k$ and pixel $i$, i.e. the probability that pixel $i$ is assigned to the $k$-th object. More formally, $m_{ik} = p_\theta(C_i^t = k|z_k^t, v^t)$, where $C_i^t$ is a categorical random variable and $C_i^t = k$ represents the event that pixel $i$ is assigned to the $k$-th object. This is an important property for object segmentation, as it implies that every pixel in $x^t$ must be explained by one and only one object. Together, the $M$ mixing coefficients for object $k$ (one per pixel) form a soft object segmentation mask $m_k = p_\theta(C^t = k|z_k^t, v^t)$. We assume all pixel values $x_{ik}^t$ are independent given the corresponding latent object representation $z_k^t$ and viewpoint $v^t$, and simplify computations by using a fixed variance $\sigma^2 = 0.01$ for all pixels. In practice, we split the parameters $\theta$ into two pieces, $\theta^1$ and $\theta^2$, in order to handle the viewpoint-queried neural transformation and observation-generation separately in two consecutive stages. That is, we first transform the $K$ latent object representations $\mathbf{z}^t$ w.r.t. a viewpoint $v^q$ using the function $f_{\theta^1}$, then we pass the output through a decoder $g_{\theta^2}$ in order to render a viewpoint-queried observation $x^q$. We illustrate this process in Figure 2b and Algorithm 1.

### 3.3 Inference

Equation 1 simplifies the inference problem by breaking the computation of the multi-object-multi-view posterior $p(\mathbf{z} = \{z_k\}|\{(x^t, v^t)\})$, into a recursive computation of multi-object-single-view posteriors, $p(\mathbf{z}^t|x^t, v^t, \mathbf{z}^{t-1})$. However, exact inference of $p(\mathbf{z}^t = \{z_k^t\}|x^t, v^t, \mathbf{z}^{t-1})$ is still intractable. Similar to IODINE[9], we apply iterative amortized inference[19] to approximate the

intractable target posterior. However, unlike IODINE which always initializes the prior from a standard Gaussian, the inference model of MulMON takes an approximate posterior from last iteration as the prior. Hence, we approximate the intractable posterior with $q_{\boldsymbol{\lambda}}(\mathbf{z}^t|x^t, v^t, \mathbf{z}^{t-1})$, where $\boldsymbol{\lambda} = \{\lambda_k\} = \{(\mu_k, \sigma_k)\}$ parametrizes a set of object-specific Gaussian distributions in the latent space. We denote the number of iterations for the inner loop with $L$, and each iteration is indexed by $l$. The parameter update in the iterative inference is thus:

$$z_k^{t(l)} \overset{k}{\sim} q_{\lambda_k^{(l)}}(z_k^t|x^t, v^t, z_k^{t-1})$$
$$\lambda_k^{(l+1)} \overset{k}{\leftarrow} \lambda_k^{(l)} + f_{\Phi}(z_k^{t(l)}, x^t, v^t, \mathbf{a}(\cdot)), \tag{3}$$

where the refinement function $f_{\Phi}$, with trainable parameter $\Phi$, is modeled by a recurrent neural network. The $\overset{k}{\sim}$ and $\overset{k}{\leftarrow}$ operators denote parallel operations over $K$ independent object slots. The same auxiliary inputs such as mask gradients $\nabla_{m_k}\mathcal{L}$ and posterior gradient $\nabla_{\lambda_k}\mathcal{L}$, where $\mathcal{L}$ is the objective function of MulMON (will be discussed in Section 3.4), as that of IODINE are also adopted to refine the posterior. These auxiliary inputs are computed by a function $\mathbf{a}(\cdot)$, namely the "auxiliary function", which takes in the refinement function's inputs along with the posterior parameter $\lambda_k^{(l)}$.

---

**Algorithm 1: MulMON at Test Time: Online Scene Learning**

---

**Input:** Trained parameters $\Phi, \theta$
**Hyperparameters** $K$, $\sigma^2 = 0.01$, $L$ **Initialize** $\boldsymbol{\lambda}^0 = \{\lambda_k^0\} \leftarrow \{(\mu_k = \mathbf{0}, \sigma_k = \mathbf{I})\}$;
/* The outer loop for scene learning                                    */
**for** $t = 1$ *to* $T$ **do**
    **Access** *a scene observation* $(x^t, v^t)$;
    $\boldsymbol{\lambda}^{prior} = \boldsymbol{\lambda}^{t(0)} \leftarrow \boldsymbol{\lambda}^{t-1}$;
    /* The inner loop for observation aggregation                      */
    **for** $l = 0$ *to* $L - 1$ **do**
        $\mathbf{z}^{t(l)} \sim \mathcal{N}(\mathbf{z}^{t(l)}; \boldsymbol{\lambda}^{t(l)})$;
        $\{\mu_{xk}^{(l)}, \hat{m}_{xk}^{(l)}\} \leftarrow g_{\theta^2}(f_{\theta^1}(\mathbf{z}^{t(l)}, v^t))$;
        $\{m_k^{(l)}\} \leftarrow \mathbf{softmax}(\{\hat{m}_{xk}^{(l)}\})$;
        /* The spatial Gaussian mixture                                */
        $p_{\theta}(x^t|\mathbf{z}^{t(l)}, v^t) \leftarrow \sum_k m_k^{(l)}\mathcal{N}(x_k^t; \mu_{xk}^{(l)}, \sigma^2\mathbf{I})$;
        **if** $l == 0$ **then**
            $\mathcal{L}_{\mathcal{T}}^{(l)} \leftarrow -\log p_{\theta}(x^t|\mathbf{z}^{t(l)}, v^t)$;
        **else**
            $\mathcal{L}_{\mathcal{T}}^{(l)} \leftarrow \mathcal{D}_{\mathrm{KL}}[\mathcal{N}(\mathbf{z}^{t(l)}; \boldsymbol{\lambda}^{t(l)})||\mathcal{N}(\mathbf{z}^{t(l)}; \boldsymbol{\lambda}^{prior})] - \log p_{\theta}(x^t|\mathbf{z}^{t(l)}, v^t)$;
        $\boldsymbol{\lambda}^{t(l+1)} \leftarrow \boldsymbol{\lambda}^{t(l)} + f_{\Phi}(z_k^{t(l)}, x^t, v^t, \mathbf{a}(\cdot))$;
    $\boldsymbol{\lambda}^t \leftarrow \boldsymbol{\lambda}^{t(l+1)}$;

---

The variational approximate posterior of MulMON is thus:

$$q_{\boldsymbol{\lambda}}(\mathbf{z} = \{z_k\}|x^{1:T}, v^{1:T}) = q(\mathbf{z}^0)\prod_{t=1}^{T} q_{\boldsymbol{\lambda}}(\mathbf{z}^t|x^t, v^t, \mathbf{z}^{t-1}), \tag{4}$$

where the initial guess $q(\mathbf{z}^0)$ is a standard Gaussian and this is the same as $p(\mathbf{z}^0)$ in equation 1. We refer to Algorithm 1 for more details about MulMON's inference process and its behaviors at test time.

## 3.4 Training

Similar to IODINE, MulMON learns the decoder parameters $\theta$ and the refinement network parameters $\Phi$ by minimizing $\mathcal{D}_{\mathrm{KL}}[q_{\boldsymbol{\lambda}}(\mathbf{z}|x^{1:T}, v^{1:T})||p_{\theta}(\mathbf{z}|x^{1:T}, v^{1:T})]$, which is equivalent to maximizing the evidence lower bound (i.e. the ELBO, denoted as $\mathcal{L}$) in a generative configuration. However, instead of directly maximizing the ELBO like IODINE, we simulate novel viewpoint-queried generation in

the training process (similar to GQN [8]). By asking MulMON to predict the appearance of a scene from unobserved viewpoints during training, we ensure that the iterative updates are indeed used to aggregate spatial information across views, as a complete 3D scene understanding is required to perform well. More formally, we randomly partition the set of $T$ scene observations $\{(x^t, v^t)\}$ into two subsets $\mathcal{T}$ and $\mathcal{Q}$, with $n \sim \mathcal{U}(1, 5)$ observations in $\mathcal{T}$ and the remaining $T - n$ observations in $\mathcal{Q}$. We perform scene learning on $\mathcal{T}$ and novel viewpoint-queried generation on $\mathcal{Q}$. We thus derive the MulMON ELBO (for one scene sample) as:

$$\mathcal{L} = \frac{1}{|\mathcal{T}|} \sum_{t \in \mathcal{T}} \mathbf{E}_{q_{\boldsymbol{\lambda}}(\mathbf{z}^t|\cdot)}[\log p_\theta(x^t|\mathbf{z}^t, v^t)] + \frac{1}{|\mathcal{Q}|} \sum_{q \in \mathcal{Q}, \, t \sim \mathcal{T}} \mathbf{E}_{q_{\boldsymbol{\lambda}}(\mathbf{z}^t|\cdot)}[\log p_\theta(x^q|\mathbf{z}^t, v^q)]$$
$$- \frac{1}{|\mathcal{T}|} \sum_{t \in \mathcal{T}} \mathbf{IG}(\mathbf{z}^t, x^t; \, v^t, \mathbf{z}^{t-1}) \tag{5}$$

where $\mathbf{IG}$ is the information gain (aka. Bayesian surprise), the operation $|\cdot|$ measures the size of a discrete set, and $q_{\boldsymbol{\lambda}}(\boldsymbol{z}^t|\cdot)$ is an abbreviation of the variational posterior $q_{\boldsymbol{\lambda}}(\mathbf{z}^t|x^t, v^t, \mathbf{z}^{t-1})$, from which we sample $\mathbf{z}^t$ by applying ancestral sampling. In practice, we use an efficient approximation of the information gain, i.e. an approximate $\mathbf{IG}$ $\mathcal{D}_{\mathrm{KL}}[q_{\boldsymbol{\lambda}}(\mathbf{z}^t|x^t, v^t, \mathbf{z}^{t-1})||q_{\boldsymbol{\lambda}}(\mathbf{z}^t|v^t, \mathbf{z}^{t-1})]$ or $\mathcal{D}_{\mathrm{KL}}[q_{\boldsymbol{\lambda}}(\mathbf{z} = \mathbf{z}^t|x^t, v^t, x^{1:t-1}, v^{1:t-1})||q_{\boldsymbol{\lambda}}(\mathbf{z} = \mathbf{z}^{t-1}|x^{1:t-1}, v^{1:t-1})]$. Note that using a fixed number of observations could harm the model's robustness at test time, hence why we randomly partition the observations into size-varying sets $\mathcal{T}$ and $\mathcal{Q}$ during training, i.e. we train the model with varying number of observations. See Appendix A for full details of the training algorithm of MulMON.

# 4 Related Work

**Single-object-single-view (SOSV).** Many recent breakthroughs in unsupervised representation learning of have come in the form of "disentanglement" models [11, 6, 14] that seek feature-attribute-level understanding by encouraging e.g. independence among latent dimensions. However, most of these models focus on a single view of a single object that has been placed in front of some background (e.g. dSprites, CelebA, 3D Chairs). As a result, they fail to i) generalize to more realistic, multi-object scenes, and ii) accurately capture 3D scene information (e.g. resolve single-view spatial ambiguities and estimate e.g. rotation along the vertical axis).

**Multi-object-single-view (MOSV).** To avoid the additional computational complexities of factorising or segmenting the objects in a scene into explicit *multi-object* representations, many works have used pre-segmented images [25, 26]. However, this comes at the cost of decreased representational power (good object representation requires good object segmentation [9]) and a reliance on annotated data. In addition, these works struggle in a multi-view scenario where pre-segmented images require consistent multi-frame object registration and tracking, since the segmentation and representation models work independently. More recently, several works [4, 9, 18, 17, 1] have succeeded in approximating the factorized posterior $p(z_1, z_2, ..., z_K|x)$ within the VAE framework, achieving impressive unsupervised object-level scene factorization. However, being single-view models, they fall victim to single-view spatial ambiguities. As a result, they fail to accurately capture the scene's 3D spatial structure, causing problems for object-level segmentation. To overcome this and learn object-based representations that accurately capture 3D spatial structures, MulMON essentially extends these models to the multi-view scenario.

**Single-object-multi-view (SOMV).** Recently, unsupervised models like GQN [8] and EGQN [22] have been quite successful in aggregating multiple observations of a scene into a single-slot representation that accurately captures the spatial layout of the 3D scene, as shown by their ability to predict the appearance of a scene from unobserved viewpoints. However, being single-slot or "single-object" models, they fail to achieve object-level scene understanding in multi-object scenes, and as a result, miss out on the aforementioned benefits of object-centric scene representations. To overcome this, MulMON essentially extends these models to the case of multi-object representations. In addition, several works have sought explicit 3D representations either in the latent space [21] or output space [24, 2]. However, due to the complexity of 1) working with explicit 3D object representations and 2) maintaining object correspondences across views, these works have been limited to single-object scenes (often quite simple, with "floating" objects placed in front of a plain background).

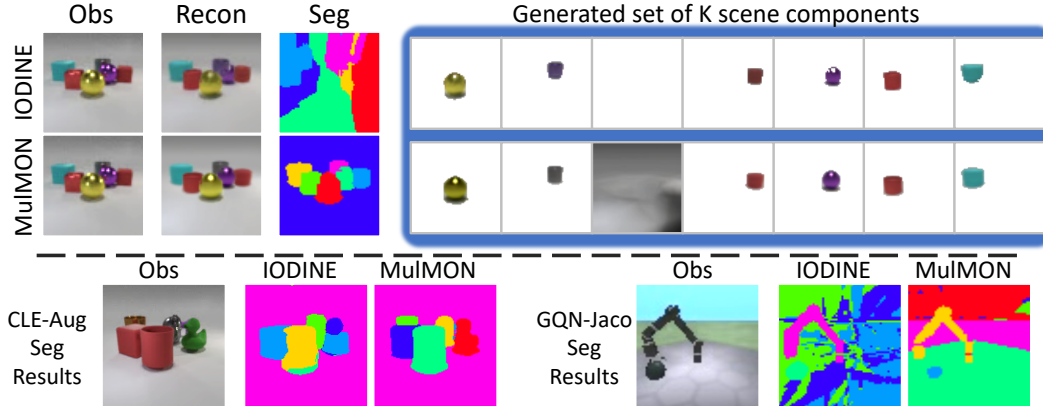

Figure 3: Qualitative comparison of MulMON vs. IODINE in terms of scene segmentation performance. **Top**: L) Reconstruction and segmentation comparison on a CLE-MV data sample. R) Individual masked-object generation using each object's representation independently (see Appendix C for the computation details). **Bottom**: Segmentation performance on CLE-Aug and GQN-Jaco data samples (specific colors arbitrary).

**Multi-object scenes in videos.** While some works in multi-object discovery and tracking in videos appear to be MOMV models [16, 12], they in fact work with one view per scene (abiding strictly by our definition of a scene in Section 2) and are only capable of dealing with binarizable MNIST-like images.

## 5   Experiments

Our experiments are designed to demonstrate that MulMON is a feasible solution to the MOMV problem, and to demonstrate that MulMON learns better representations than the MOSV and SOMV models by resolving spatial ambiguity. To do so, we compare the performance of MulMON against two baseline models, IODINE[9] (MOSV) and GQN[8] (SOMV), in terms of segmentation, viewpoint-queried prediction (appearance and segmentation) and disentanglement (inter- and intra-object). To best facilitate these comparisons, we created two new datasets called CLEVR6-MultiView (abbr. CLE-MV) and CLEVR6-Augmented (abbr. CLE-Aug) which contain ground-truth segmentation masks and shape descriptions (e.g. colors, materials, etc.). The CLE-MV dataset is a multi-view, observation-enabled variant (10 views per scene) of the CLEVR6 dataset[13, 9]. The CLE-Aug adds more complex shapes (e.g. horses, ducks, and teapots etc.) to the CLE-MV environment. In addition, we compare the models on the GQN-Jaco dataset[8] and use the GQN-Shepard-Metzler7 dataset[8] (abbr. Shep7) for a specific ablation study. We train all models using an Adam optimizer with an initial learning rate $0.0003$ for $300k$ gradient steps. In addition, all experiments were run across five different random seeds to simulate scenarios of different observation orders and view selections. For more details about the four datasets and model implementations see Appendix B and C respectively in the supplementary materials.

### 5.1   Scene Factorization

The ability of MulMON to perform scene object decomposition in the scene learning phase is crucial for learning object-centric scene representations. We evaluate its segmentation ability by computing mean-intersection-over-union (mIoU) scores between the output and the GT masks. However, since the segments produced by IODINE and MulMON are unordered, GT masks and object segmentation masks need to first be one-to-one registered for each scene. We solve this matching problem by first computing every possible object pairs of GT object masks and outputs, then, for each GT object mask, we find the output object mask that gives the highest IoU score. Table 1a shows that MulMON outperforms IODINE in object segmentation. The qualitative comparison in Figure 3 shows that IODINE captures each object well independently but fails to understand the spatial structure along depth directions (3D) – as described by the Categorical distribution (see Section 3.2). Note that IODINE's poor segmentation performance is mostly due to its poor handling of the background, i.e. its tendency to split up the background. Although the background is often considered a less-important

| Models | CLE-MV | CLE-Aug |
|---|---|---|
| GQN | N/A | N/A |
| IODINE | $0.1891 \pm 0.0000$ | $0.5137 \pm 0.0007$ |
| MulMON | $\mathbf{0.7852 \pm 0.0008}$ | $\mathbf{0.7076 \pm 0.0004}$ |

(a) Object Segmentation mIoU Scores

| Models | CLE-MV | CLE-Aug |
|---|---|---|
| GQN | N/A | N/A |
| IODINE | N/A | N/A |
| MulMON | $\mathbf{0.7845 \pm 0.0011}$ | $\mathbf{0.6860 \pm 0.0006}$ |

(b) Predicted Object Segmentation mIoU Scores

| Models | CLE-MV | CLE-Aug | GQN-Jaco |
|---|---|---|---|
| GQN | $0.1426 \pm 0.0002$ | $0.1482 \pm 0.0001$ | $0.1675 \pm 0.0013$ |
| IODINE | N/A | N/A | N/A |
| MulMON | $\mathbf{0.0464 \pm 0.0004}$ | $\mathbf{0.0733 \pm 0.0003}$ | $\mathbf{0.1607 \pm 0.0018}$ |

(c) Predicted Observation RMSE (pixel avg.)

| Models | Disent. | Compl. | Inform. |
|---|---|---|---|
| GQN | N/A | N/A | N/A |
| IODINE | $0.47 \pm 0.00$ | $0.60 \pm 0.01$ | $0.67 \pm 0.01$ |
| MulMON | $\mathbf{0.65 \pm 0.01}$ | $\mathbf{0.73 \pm 0.01}$ | $\mathbf{0.78 \pm 0.00}$ |

(d) Disentanglement Analysis (CLE-MV)

Table 1: Quantitative comparisons of MulMON, IODINE and GQN. "N/A" denotes cases where a model is *unable* to perform a task. In tables (a), (b) and (d), higher is better and 1 is best. For table (c), lower is better and 0 is best.

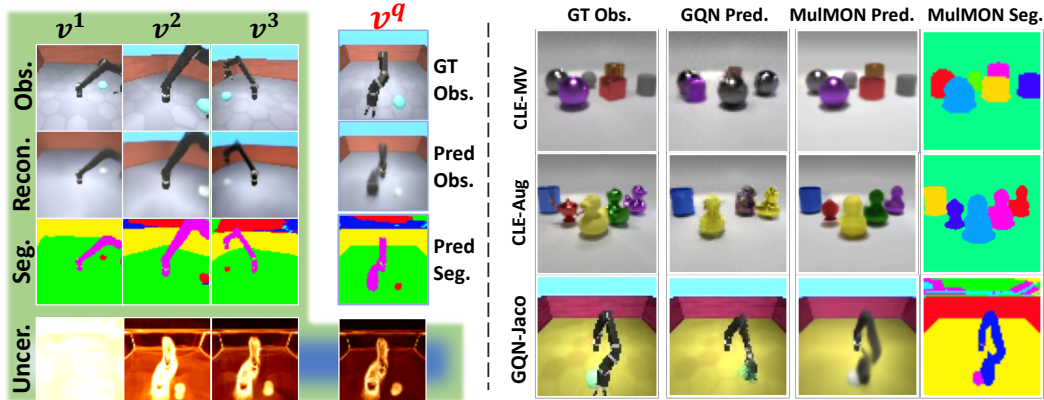

Figure 4: Qualitative results on novel-viewpoint prediction. **Left**: Working example of MulMON, including uncertainty reduction across views, for a GQN-Jaco sample. **Right**: Qualitative comparison between MulMON and GQN.

"object", correct handling of the background demonstrates better spatial-reasoning ability. Together, all of these results suggest that MulMON learns better single-object representations and spatial structures by overcoming spatial ambiguities.

## 5.2 Novel-viewpoint Prediction

MulMON can predict both observations and segmentation for novel viewpoints. This is the major advantage of our model (a MOMV model) over the MOSV and SOMV models in scene understanding. For our evaluation of online scene learning, each model is provided with 5 observations of each scene and then asked to predict both the observation and segmentation for randomly-selected novel viewpoints. We compute the root-mean-square error (RMSE) and mIoU as quality measures of the predicted observation and segmentation respectively. Table 1c shows that MulMON outperforms GQN on novel-view observation prediction. Table 1b shows that MulMON is the only model that can predict the object segmentation for novel viewpoints – and it does so with a similar quality to the original object segmentation (compare with Table 1a). However, as shown in Figure 4, GQN tends to capture more pixel details than MulMON, albeit at the risk of predicting wrong spatial configurations.

## 5.3 Disentanglement Analysis

To evaluate how well MulMON performs disentanglement at both the inter-object level and the intra-object level, we run disentanglement analyses on the representations learned by MulMON. For our qualitative analysis, we pick one of $K$ objects in a scene, and traverse one dimension of the learned object-representation at a time. Figure 5 (left) shows i) MulMON's intra-object disentanglement, encoding interpretable features in different latent dimensions; and ii) MulMON's inter-object disentanglement, allowing single-object manipulation without affecting other objects in

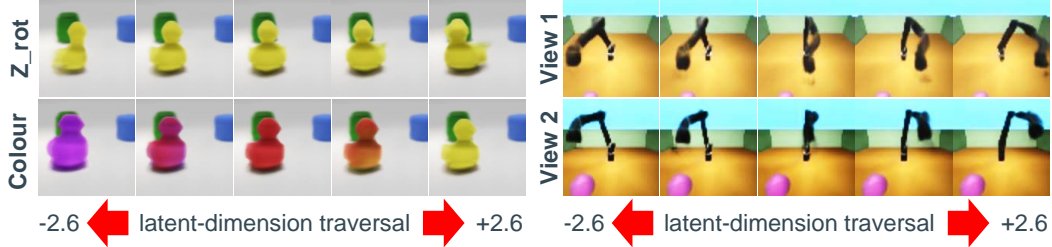

-2.6 ← latent-dimension traversal → +2.6     -2.6 ← latent-dimension traversal → +2.6

Figure 5: Single-object manipulations via latent traversals. **Left** Traversing two dimensions of the duck's latent representation (one per row). Top row cropped for visual clarity. **Right** For two different views (one per row), we manipulate the dimension of the learned representation that appears to capture vertical-axis rotation.

the scene. Figure 5 (right) shows that MulMON captures 3D information (vertical-axis rotation) and broadcasts consistent manipulations of this 3D information to different views. For our quantitative analysis, we employ the method of Eastwood and Williams[7] to compare the representations learned by each model on the CLE-MV and CLE-Aug datasets. As shown in Table 1d, MulMON learns object representations that are more disentangled, complete (compact) and informative (about ground-truth object properties). See Appendix D for further details.

## 5.4 Ablation Study

We consider the number of observations $T$ to be the most important hyperparameter of MulMON as the key insight of MulMON is to reduce multi-object spatial uncertainty by aggregating information across multiple observations. To visualize the effect of $T$ on MulMON's performance, we plot MulMON's uncertainty about the scene as a function of $T$. More specifically, for a given scene and ordering of the observations, we: 1) draw 10 samples from the approximate variational posterior $q_{\boldsymbol{\lambda}}(\mathbf{z} = \{z_k\}|x^{1:T}, v^{1:T})$, 2) obtain the corresponding viewpoint-queried observation predictions using the 10 latent samples (see Section 3.2 and Figure 2b), 3) compute the pixel-wise empirical variance over these observation predictions. Averaging over all scenes in the dataset and sampling

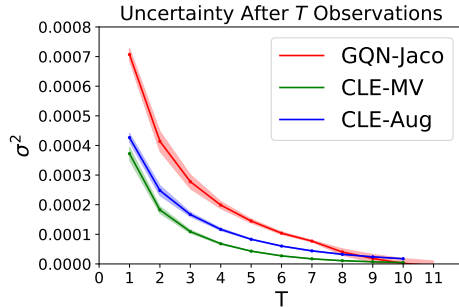

Figure 6: Uncertainty vs. $T$.

5 random view orderings (5 different random seeds), we can then create Figure 6 which shows that MulMON effectively reduces the spatial uncertainty/ambiguity by leveraging multiple views. In particular, MulMON's uncertainty is rapidly reduced after only a small number of observations $T$. We also study the effects of two other important hyperparameters, namely the globally-fixed number of object slots $K$ and the coefficient of information gain **IG** (in the MulMON ELBO). For details on these further ablation studies, we refer the reader to Appendix D.

## 6 Conclusion

We have presented MulMON—a method for learning accurate, object-centric representations of multi-object scenes by leveraging multiple views. We have shown that MulMON's ability to aggregate information across multiple views does indeed allow it to better-resolve spatial ambiguity (or uncertainty) and better-capture 3D spatial structures, and as a result, outperform state-of-the-art models for unsupervised object segmentation. We have also shown that, by virtue of addressing the more complicated multi-object-multi-view scenario, MulMON achieves new functionality—the prediction of both appearance and object segmentations for novel viewpoints. As all scenes in this paper are static, future work may look to extend MulMON to dynamic multi-object scenes.

## Acknowledgments and Disclosure of Funding

This research is partly supported by the *Trimbot2020* project, which is funded by the European Union Horizon 2020 programme. The authors would like to thank Prof. C. K. I. Williams for his valuable advice that helps to improve this work and acknowledge the GPU computing support from Dr. Zhibin Li's Advanced Intelligence Robotics Lab at the University of Edinburgh.

## Broader Impact

In this paper, we presented a new method to learn object-centric representations of multi-object scenes. Object-centric scene representations can support many downstream tasks, such as autonomous scene exploration, object segmentation (tracking) and scene synthesizing.

Autonomous scene exploration has real-world applications in exploring hazardous environments, mines, potential bomb threats, nuclear waste zones. This could have societal impacts through increased worker safety or potential military (mis)uses.

Object detection and tracking has real-world applications in tracking people in CCTV footage, detecting buildings from aerial footage, and spotting potential hazards for autonomous vehicles. Potential societal impacts include safer autonomous vehicles and unwanted/increased surveillance.

Finally, scene synthesizing has applications in automated scene modelling for computer games. This further transitions society away from labor-intensive tasks to higher-level cognitive tasks. This could have both positive (more time for cognitive tasks) and negative (less employment) impacts on society.

## Footnotes

[1]Code available at https://github.com/NanboLi/MulMON

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
