[Supplementary Material]


**Note: we use the same notations in the Appendix as that in the main paper.**

## A. Training Algorithm of MulMON

We refer to Algorithm 1 and 2 for the training algorithm of MulMON.

---
**Algorithm 1: MulMON Training Algorithm**

---
**Data** *a set of $N$ scenes as $\{(images\ x^{1:T}, viewpoints\ v^{1:T})\}_N$*
**begin**

    **Initialize** *trainable parameters* $\Phi^{(0)}$, $\theta^{(0)}$, step count $s = 0$;

    **repeat**

        **Sample** *mini batch* $\{(x^{1:T}, v^{1:T})\}_M \sim \{(x^{1:T}, v^{1:T})\}_N$, where $M \leq N$;

        `/* The below loop can go parallel as tensor operations    */`

        **for** $(x^{1:T}, v^{1:T})$ in $\{(x^{1:T}, v^{1:T})\}_M$ **do**

            $\mathcal{L}_m \leftarrow$ **SingleSampleELBO**$((x^{1:T}, v^{1:T}), \Phi^{(s)}, \theta^{(s)})$;

        $\mathcal{L} = \frac{1}{M} \sum_{m=1}^{M} \mathcal{L}_m$;

        `/* gradient update                                          */`

        $\Phi^{(s+1)} \leftarrow$ **optimizer**$(\mathcal{L}, \Phi^{(s)})$ ;

        $\theta^{(s+1)} \leftarrow$ **optimizer**$(\mathcal{L}, \theta^{(s)})$;

        $s \leftarrow s + 1$;

    **until** $\Phi, \theta$ *converge*;

---

## B. Data Configurations

We show samples of the used datasets in Figure 8. **CLEVR-MultiView & CLEVR-Augmented** We

Figure 1: Examples of the four dataset used in this work.

adapt the Blender environment of the original CLEVR datasets[4] to render both datasets. We make a scene by randomly sampling $3 \sim 6$ rigid shapes as well as their properties like poses, materials, colors etc.. For the CLEVR-MultiView (CLE-MV) dataset, we sample shapes from three categories: cubes, spheres, and cylinders, which are the same as the original CLEVR dataset. For the CLEVR-Augmented (CLE-Aug), we add more shape categories into the pool: mugs, teapots, ducks, owls, and horses. We render 10 image observations for each scene and save the 10 camera poses as 10 viewpoint vectors. We use resolution $64 \times 64$ for the CLE-MV images and $128 \times 128$ for the CLE-Aug images. All viewpoints are at the same horizontal level but different azimuth with their focuses locked at the

**Algorithm 2: SingleSampleELBO**

---

**Input:** A single scene sample (images $x^{1:T}$, viewpoints $v^{1:T}$), trainable parameters $\Phi$, $\theta$
**Hyperparameters** $K$, $\sigma^2$, $L$
**begin**

$\quad \mathcal{T} = \{(x^t, v^t)\}, \mathcal{Q} = \{(x^q, v^q)\} \xleftarrow{random\ split\ T} (x^{1:T}, v^{1:T})$ ;

$\quad$ **Initialize** $\boldsymbol{\lambda}^0 = \{\lambda_k^0\} \leftarrow \{(\mu_k = \mathbf{0}, \sigma_k = \mathbf{I})\}$;

$\quad$ /* The outer loop for scene learning $\qquad\qquad\qquad\qquad\qquad\qquad$ */

$\quad$ **for** $t = 1$ *to* $|\mathcal{T}|$ **do**

$\qquad$ **Access** *a scene observation* $(x^t, v^t)$;

$\qquad \boldsymbol{\lambda}^{prior} = \boldsymbol{\lambda}^{t(0)} \leftarrow \boldsymbol{\lambda}^{t-1}$;

$\qquad$ /* The inner loop for observation aggregation $\qquad\qquad\qquad$ */

$\qquad$ **for** $l = 0$ *to* $L - 1$ **do**

$\qquad\quad \mathbf{z}^{t(l)} \sim \mathcal{N}(\mathbf{z}^{t(l)}; \boldsymbol{\lambda}^{t(l)})$;

$\qquad\quad \{\mu_{xk}^{(l)}, \hat{m}_k^{(l)}\} \leftarrow g_{\theta^2}(f_{\theta^1}(\mathbf{z}^{t(l)}, v^t))$;

$\qquad\quad \{m_k^{(l)}\} \leftarrow \mathbf{softmax}(\{\hat{m}_k^{(l)}\})$;

$\qquad\quad p_\theta(x^t | \mathbf{z}^{t(l)}, v^t) \leftarrow \sum_k m_k^{(l)} \mathcal{N}(x_k^t; \mu_{xk}^{(l)}, \sigma^2 \mathbf{I})$;

$\qquad\quad$ **if** $l == 0$ **then**

$\qquad\qquad\quad \mathcal{L}_{\mathcal{T}}^{(l)} \leftarrow -\log p_\theta(x^t | \mathbf{z}^{t(l)}, v^t)$;

$\qquad\quad$ **else**

$\qquad\qquad\quad \mathcal{L}_{\mathcal{T}}^{(l)} \leftarrow \mathcal{D}_{\mathrm{KL}}[\mathcal{N}(\mathbf{z}^{t(l)}; \boldsymbol{\lambda}^{t(l)}) || \mathcal{N}(\mathbf{z}^{t(l)}; \boldsymbol{\lambda}^{prior})] - \log p_\theta(x^t | \mathbf{z}^{t(l)}, v^t)$;

$\qquad\quad \boldsymbol{\lambda}^{t(l+1)} \leftarrow \boldsymbol{\lambda}^{t(l)} + f_\Phi(z_k^{t(l)}, x^t, v^t, \mathbf{a}(\cdot))$;

$\qquad \boldsymbol{\lambda}^t \leftarrow \boldsymbol{\lambda}^{t(l+1)}$;

$\qquad \mathcal{L}_{\mathcal{T}}^t \leftarrow \frac{2l+2}{L^2+L} \sum_l \mathcal{L}_{\mathcal{T}}^{(l)}$;

$\quad$ /* Viewpoint-queried prediction $\qquad\qquad\qquad\qquad\qquad\qquad\qquad$ */

$\quad$ **for** $(x^q, v^q)$ *in* $\mathcal{Q}$ **do**

$\qquad \mathbf{z}^t \sim \mathcal{N}(\mathbf{z}^t; \boldsymbol{\lambda}^t)$;

$\qquad \{\mu_{xk}^q, \hat{m}_k^q\} \leftarrow g_{\theta^2}(f_{\theta^1}(\mathbf{z}^t, v^q))$;

$\qquad \{m_k^q\} \leftarrow \mathbf{softmax}(\{\hat{m}_k^q\})$;

$\qquad p_\theta(x^q | \mathbf{z}^t, v^q) \leftarrow \sum_k m_k \mathcal{N}(x_k^q; \mu_{xk}^q, \sigma^2 \mathbf{I})$ ;

$\qquad \mathcal{L}_{\mathcal{Q}}^q \leftarrow -\log p_\theta(x^q | \mathbf{z}^t, v^q)$;

$\quad$ /* Compute the MulMON ELBO $\qquad\qquad\qquad\qquad\qquad\qquad\qquad$ */

$\quad \mathcal{L} = \frac{1}{|\mathcal{T}|} \sum_t \mathcal{L}_{\mathcal{T}} + \frac{1}{|\mathcal{Q}|} \sum_q \mathcal{L}_{\mathcal{Q}}$

**Output:** $\mathcal{L}$

---

scene center. We thus parametrize a viewpoint 3-D viewpoint vector as $(\cos\alpha, \sin\alpha, r)$, where $\alpha$ is the azimuth angle and $r$ is the distance to the scene center. In addition, we save the object properties (e.g. shape categories, materials, and colors etc.) and generate objects' segmentation masks for quantitative evaluations. CLEVR-MultiView (CLE-MV) contains 1500 training scenes, 200 testing images. CLEVR-Augmented (CLE-Aug) contains 2434 training scenes and 500 testing scenes.

**GQN-Jaco** We use a mini subset of the original GQN-Jaco dataset[2] in our paper. The original GQN-Jaco contains 4 million scenes, each of them contains 20 image observations (resolution: $64 \times 64$) and 20 corresponding viewpoint vectors (7D). To reduce the storage memory and accelerate the training, we randomly sample $2,000$ scenes for training and 500 scenes for testing. Also, for each scene, we use only 11 observations (viewpoints) that are randomly sampled from the 20 observations of the original dataset.

**GQN-Shepard-Metzler7** Same as the GQN-Jaco dataset, we make a mini GQN-Shepard-Metzler7 dataset[2] (Shep7) by randomly selecting 3000 scenes for training and 200 for testing. Each scene contains 15 images observations (resolution: $64 \times 64$) with 15 corresponding viewpoint vectors (7D). We use Shep7 to study the effect of $K$ on our model.

## C. Implementation Details

**Training configurations** See Table 1 for our training configurations.

Table 1: Training Configurations

| TYPE | THE TRAININGS OF MULMON, IODINE, GQN |
|---|---|
| OPTIMIZER | ADAM |
| INITIAL LEARNING RATE $\eta_0$ | $3e^{-4}$ |
| LEARNING RATE AT STEP $s$ | $\max\{0.1\eta_0 + 0.9\eta_0 \cdot (1.0 - s/6e^5), 0.1\eta_0\}$ |
| TOTAL GRADIENT STEPS | $300k$ |
| BATCH SIZE | 8 FOR CLE-MV, CLE-AUG, 16 FOR GQN-JACO, 12 FOR SHEP7 |
| * THE SAME SCHEDULER AS THE ORIGINAL GQN EXCEPT FOR FASTER ATTENUATION | |

Table 2: Model State Space Specifications

| TYPE | CLE-MV | CLE-AUG | GQN-JACO | SHEP7 |
|---|---|---|---|---|
| Z_DIMS | 16 | 16 | 32 | 16 |
| V_DIMS | 3 | 3 | 7 | 7 |
| Z_DIMS: THE DIMENSION OF A LATENT REPRESENTATION | | | | |
| V_DIMS: THE DIMENSION OF A VIEWPOINT VECTOR | | | | |

**Model architecture** We show our model configurations in Table 2, 3, and 4.

Table 3: MulMON Refinement Network with Trainable Parameters $\Phi$

| Parameters | Type | Channels (out) | Activations. | Descriptions |
|---|---|---|---|---|
| | Input | 17 | | * Auxiliary inputs $\mathbf{a}(x^t)$ |
| $\Phi$ | Conv $3 \times 3$ | 32 | Relu | |
| | Conv $3 \times 3$ | 32 | Relu | |
| | Conv $3 \times 3$ | 64 | Relu | |
| | Conv $3 \times 3$ | 64 | Relu | |
| | Flatten | | | |
| | Linear | 256 | Relu | |
| | Linear | 128 | Linear | |
| | Concat | 128+4*z_dims | | |
| | LSTMCell | 128 | | |
| | Linear | 128 | Linear | output $\Delta\lambda$ |

z_dims: the dimension of a latent representation
v_dims: the dimension of a viewpoint vector
Stride= 1 set for all Convs.
* see IODINE[3] for details
LSTMCell channels: the dimensions of the hidden states

**Decoder-output processing** For a single view of a scene, our decoder $g_\theta$ outputs K $3 \times H \times W$ RGB values (i.e. $\{x_k\}$ as in equation 2 of the main paper) along with K $1 \times H \times W$ mask logits (denoted as $\{\hat{m}_k\}$). $H$ and $W$ are the same as the image sizes, i.e. height and width. In this section, we detail the computation of rendering K individual scene components' images, segmentation masks, and reconstructed scene images. We compute the individual scene objects' images as:

$$x_k \overset{k}{\leftarrow} \mathbf{sigmoid}(\hat{m}_k) \cdot x_k.$$

As shown in Figure 2, this overcomes mutual occlusions of the objects since the **sigmoid** functions do not impose any dependence on K objects. We compute the segmentation masks as:

$$m_k \overset{k}{\leftarrow} \mathbf{softmax_k}(\hat{m}_k).$$

Table 4: MulMON Decoder with Trainable Parameters $\theta$

| Parameters | Type | Channels (out) | Activations. | Descriptions |
|---|---|---|---|---|
| | Input | z_dims+ v_dims | | $z_k \sim \mathcal{N}(z_k; \lambda_k), v$ |
| $\theta^1$ (view transformer) | Linear | 512 | Relu | |
| | Linear | z_dims | Linear | $\tilde{z}_k = f_{\theta_1}(z_k, v)$ |
| | Input | z_dims | | $\tilde{z}_k = f_{\theta_1}(z_k, v)$ |
| $\theta^2$ (Generator) | Broadcast | z_dims+2 | | * Broadcast to grid |
| | Conv $3 \times 3$ | 32 | Relu | |
| | Conv $3 \times 3$ | 32 | Relu | |
| | Conv $3 \times 3$ | 32 | Relu | |
| | Conv $3 \times 3$ | 32 | Relu | |
| | Conv $3 \times 3$ | 4 | Linear | rgb mean ($\mu_{xk}$) + mask logits ($\hat{m}_k$) |

z_dims: the dimension of a latent representation
v_dims: the dimension of a viewpoint vector
*: see spatial broadcast decoder [5]
Stride= 1 set for all Convs.

**Obs** **Rec/Pred** **Generated K Scene Components** **Seg**

Figure 2: Example of images of post-processed decoder outputs: (left to right) (predicted) scene reconstruction, generated K individual scene components (white background for visual clarity), segmentation map. The generated scene components overcome/impute occlusions (e.g. the purple glossy sphere).

To generate binary segmentation masks, we take **argmax** operation over the K $\hat{m}$ at every pixel location and encode the maximum indicator (indices) using one-hot codes. We render a scene image using a composition of all scene objects as:

$$x = \sum_k \mathbf{softmax_k}(\hat{m}_k) \cdot x_k$$
$$= \sum_k m_k \cdot x_k.$$

# D.  Additional Results

| Models | Disent. | Compl. | Inform. |
|---|---|---|---|
| GQN | N/A | N/A | N/A |
| IODINE | 0.54 | 0.48 | 0.21 |
| MulMON | **0.63** | **0.54** | **0.58** |

Table 5: Disentanglement Analysis (CLE-Aug)

## D.1  Disentanglement Analysis

To compare quantitatively the intra-object disentanglement achieved by MulMON and IODINE, we employ the framework and metrics (DCI) of Eastwood and Williams[1]. Specifically, let **y** be the ground-truth generative factors for a single-view of a single object in a single scene, and let **z** be the corresponding learned representation. Following [1], we learn a mapping from $Z = (\mathbf{z}_1, \mathbf{z}_2, \ldots)$ to $Y = (\mathbf{y}_1, \mathbf{y}_2, \ldots)$ with random forests in order to quantify the disentanglement, completeness and informativeness of the learned object representations. Section 5.3 presented the results on the CLE-MV dataset, and here we present the results on the CLE-Aug dataset. As shown in Table 5, MulMON again outperforms IODINE, learning representations that are more disentangled,

complete and informative (about ground-truth factor values). It is worth noting the significant gap in informativeness $(1 - NRMSE)$ in Table 5. This strongly indicates that the object representations learned by MulMON are more accurate, i.e. they better-capture object properties.

## D.2 Generalization Results

To evaluate MulMON's generalization ability, we trained MulMON, IODINE and GQN on CLE-Aug. Then, we compared their performance on CLE-MV and 2 new datasets—Black-Aug and UnseenShape (see Figure 3). Black-Aug contains the CLE-Aug objects but only in single, unseen colour (black). This tests the models' ability perform segmentation without colour differences/cues. UnseenShape contains only novel objects that are not in the CLE-Aug dataset—cups, cars, spheres, and diamonds. This directly tests generalization capabilities. Both datasets contain 30 scenes, each with 10 views. Table 1 shows that 1) all models generalize well to novel scenes, 2) MulMON still performs best for all tasks but observation prediction— where GQN does slightly better due to its more direct prediction procedure (features $\rightarrow$ layout vs. features $\rightarrow$ objects $\rightarrow$ layout), 3) MulMON can indeed understand the composition of novel objects in novel scenes— impressive novel-view predictions (observations and seg-

Figure 3: Samples from novel-scene datasets.

mentations) and disentanglement. Despite the excellent quantitative performance achieved by MulMON in generalization, we discovered that MulMON tended to decompose some objects, e.g. cars, into pieces (see Figure 4). Future investigations are thus needed in order to enable MulMON to generalize to more complex objects.

Table 6: MulMON's generalization performance.

| Tasks | Models | CLE-Aug (train) | CLE-MV | Black-Aug | UnseenShape |
|---|---|---|---|---|---|
| Seg. (mIoU) | IODINE | $0.51 \pm 0.001$ | $0.61 \pm 0.002$ | $0.50 \pm 0.006$ | $0.51 \pm 0.004$ |
| | MulMON | $\mathbf{0.71 \pm 0.000}$ | $\mathbf{0.71 \pm 0.004}$ | $\mathbf{0.67 \pm 0.002}$ | $\mathbf{0.64 \pm 0.004}$ |
| Pred.Obs (RMSE) | GQN | $0.15 \pm 0.000$ | $\mathbf{0.15 \pm 0.001}$ | $\mathbf{0.24 \pm 0.003}$ | $\mathbf{0.17 \pm 0.002}$ |
| | MulMON | $\mathbf{0.07 \pm 0.000}$ | $0.16 \pm 0.002$ | $0.26 \pm 0.002$ | $0.21 \pm 0.006$ |
| Disent. (D,C,I) | IODINE | $0.54, 0.48, 0.21$ | $0.14, 0.12, 0.26$ | $0.2, 0.26, 0.27$ | $0.13, 0.12, 0.26$ |
| | MulMON | $\mathbf{0.63, 0.54, 0.68}$ | $\mathbf{0.52, 0.48, 0.63}$ | $\mathbf{0.55, 0.55, 0.66}$ | $\mathbf{0.5, 0.47, 0.67}$ |
| Pred.Seg (mIoU) | MulMON | $\mathbf{0.69 \pm 0.001}$ | $\mathbf{0.71 \pm 0.004}$ | $\mathbf{0.68 \pm 0.005}$ | $\mathbf{0.60 \pm 0.005}$ |

## D.3 Ablation Study

**Prediction performance vs. the number of observations T** In Section 5.4 of the main paper, we show that the spatial uncertainty MulMON learns decrease as more observations $(T)$ are acquired. Here we study the effect of $T$ on MulMON's task performance, i.e. novel-viewpoint prediction. We employ mIoU (mean intersection-over-union) and RMSE (root-mean-square error) to measure MulMON's performance on observation prediction and segmentation prediction respectively. In Figure 5, we show that the spatial uncertainty reduction (Left) suggests boosts of task performance (Right). This means MulMON does leverage multi-view exploration to learn more accurate scene representations (than a single-view configuration), this also explains the performance gap (see Section 5.1 in the main paper) between IODINE and MulMON. To further demonstrate the advantage that MulMON has over both IODINE and GQN, we compare their performance in terms of both segmentation and novel-view appearance prediction, as a function of the number of observations given to the models. Figure 6 shows that; 1) MulMON significantly outperforms IODINE even with a single view, likely due to a superior 3D scene understanding gained during training (figures on the left), 2) Despite the more difficult task of achieving object-level segmentation, MulMON closely mirrors the performance of GQN in predicting the appearance of the scene from unobserved viewpoints (figures on the right), 3) MulMON achieves similar performance in scene segmentation

Figure 4: Failure cases of MulMON in generalization: splitting an car into pieces. Here the MulMON is trained on the CLE-MV dataset and test on the UnseenShape data.

Figure 5: Effect of T: as more observations are acquired, (Top) the spatial uncertainty reduces and the performance of novel-viewpoint prediction on observation (Bottom left) and segmentation (Bottom right) prediction boosts.

from observed and unobserved viewpoints, with any difference diminishes as the number of views increase (see dashed lines vs. solid lines in the left-hand figures).

**Effect the number of object slots K** Although explicit assumption about the number of objects in a scene is not required for MulMON, selecting a appropriate $K$ (i.e. the number of object slots) is crucial to have MulMON work correctly. In the main paper, we discussed that "$K$ needs to be sufficiently larger than the number of scene objects" and we show the experimental support here. We train our model on CLE-MV, where each scene contains 4 to 7 objects including the background, using $K = 9$ and run tests on novel-viewpoint prediction tasks using various $K$. Figure 7 shows

Figure 6: Performance comparison w.r.t. a different number of observations $T$. (Top left) Segmentation performance vs. number of observations $T$ on CLE-MV dataset. Note that "obs" means that MulMON reconstructs the observed images (scene appearances) and "unobs" means that MulMON predicts the appearance of the scene from unoberserved viewpoints. (Top right) Novel-viewpoint oberservation prediction performance vs. number of observations given to the models on CLE-MV dataset. (Bottom left) Segmentation performance vs. number of observations $T$ on CLE-MV dataset. (Bottom right) RMSE of appearance predictions for unobserved viewpoints vs. number of observations on CLE-Aug dataset.

that, for both observation prediction and segmentation prediction tasks, the model's performance improves as $K$ increases until reaching a critical point at $K = 7$, which is the maximum quantity of scene objects in the dataset. Therefore, one should select a $K$ that is always greater or equal to the maximum number of objects in a scene. When this condition is satisfied, further increase $K$ will mostly not affect MulMON's performance.

Subtle cases in terms of $K$'s selection do exist. As shown in Figure, instead of treating the Shep7 scene a combination of a single object and the background, MulMON performs as a part segmenter that discovers the small cubes and their spatial composition. This is because, in the training phase of MulMON, the amortized parameters $\Phi$ and $\theta$ are trained to capture the object features (possibly disentangled) shared across all the objects in the whole dataset instead of each scene with specific objects. These shared object features are what drives the segmentation process of MulMON. In Shep7, what is being shared are the cubes, the object itself is a spatial composition of the cubes. The results on Shep7 along with the results shown in Figure 4 illustrate the subjectiveness of perceptual grouping and leaves much space for us to study in the future.

**Effect of the IG coefficient in scene learning** In the MulMON ELBO, we fix the coefficient of the information gain at 1. In testing, we consider this coefficient controls the scene learning rate (uncertainty reduction rate). We denote the coefficient as $\alpha_{\mathbf{IG}}$ hereafter. According to the MulMON ELBO (to maximize), the negative sign of the **IG** term suggests that greater value of the coefficient

Figure 7: Effect of K: (Left) the spatial uncertainty decreases and (Middle & Right) the task performance (novel-viewpoint prediction) boosts.

Figure 8: MulMON on Shep7. MulMON treats an Shep7 object as composition of parts (cubes) instead of one object.

leads to less information gain (spatial exploration). To verify this, we try four different $\alpha_{\mathbf{IG}}$ (0.1, 1.0, 10.0 and 100.0) and track the prediction uncertainty as observations are acquired (same as our ablation study of $T$). The results in Figure 9 verifies our assumption the scene learning rate: larger $\alpha_{\mathbf{IG}}$ leads to slower scene learning and vice versa.

## D.4  Random Scene Generation

As a generative model, MulMON can generate random scenes by composing independently-sampled objects. However, to focus on forming accurate, disentangled representations of multi-object scenes, we must assume objects are i.i.d. and thus ignore inter-object correlations—e.g. two objects can appear at the same location. Figure 10 shows some random scene examples generated by MulMON (trained on the CLE-MV dataset). We can see that MulMON generates mostly good object samples by randomly composing different features but does not take into account the objects' spatial correlations.

Figure 9: Scene learning rate vs. **IG** coefficient (denoted as $\alpha_{\text{IG}}$). (Left) Uncertainty reduction gets slower when increase increase $\alpha_{\text{IG}}$. (Right) The computed uncertainty change rate or scene learning rate (lower means slower) shows larger $\alpha_{\text{IG}}$ slows the scene learning.

Figure 10: Random scene generation samples.