[Reviews · NeurIPS 2020]

Review 1

Summary and Contributions: [Main Contributions] - The proposed method is able to learn disentangled 3D object-centric scene representations for multi-object scenes in an unsupervised fashion. The inferred representations are distinct and correspond to one object. Significance: high - An iterative amortized inference method for computing P(z1, z2, …, zK | x1, x2, ... xT) efficiently. Significance: medium Reviewer’s legend for the level of significance: [very low, low, medium, high, very high] The ratings are a function of the results/analysis shown in the paper, current state of the art methods and the reviewer’s perspective on the authors’ approach [High-level summary of the paper] This paper proposes a VAE-based model to learn object-based scene representations for multiple objects in a static scene given multiple rendered views of that scene in an unsupervised fashion. Having multiple objects and multiple views makes posterior inference and maintaining object correspondence to the inferred representations difficult. The authors’ proposed approach addresses both problems. [Low-level summary] The proposed model is a variant of VAE which has a latent factorization that allows inferring multiple objects. They also get inspiration from prior work on iterative inference for VAEs and propose an inference mechanism to allow the model efficiently learn an object-centric scene representation from multiple views of a scene which contains multiple objects. During training, the model learns to infer [1,...K] objects (d-dimensional Gaussian latents) in the scene where K upper-bounds the number of objects the model can recognize, and K is set to a high enough value. During training 5 views of a scene are presented and the model is expected to reconstruct both the final rendering and object segmentations for a randomly queried novel viewpoint. They evaluate their their model on GQN-Jaco and two variant so the CLEVR datasets. They compare their model to IODINE and GQN for object segmentation, novel queried viewpoint prediction and disentanglement analysis; the results show that their method performs better quantitatively and qualitatively. They also demonstrate that their model has learned good feature-level disentangled representations.

Strengths: - The proposed method is novel and seems to work well - The experiments show that the proposed method performs better than previous best known methods - The model is able to learn disentangled object representations Technical strengths: - No 3D supervision except using unlabeled 2D images

Weaknesses: - The paper is not written well overall and is hard to read; This is specially true for the first 2 sections of the paper. The first sections of paper needs a complete revision. Throughout the paper, specially in the introduction and related works, there are lots of repetitions of definitions and sometimes relevant information are not in one place (i.e. they are a bit scattered throughout the paper). Also, there are lots of parentheses or parts of the text that seem to act as a reminder to the reader for the points the authors want the reader to remember. The technical writing needs some improvement as well; there are some important technical details of the work that are missing in the introduction and start appearing in the method section. Technical weaknesses: - For novel view prediction, the authors provide 5 views of the scene to the model which seems a lot. - It would be much more interesting if the authors could show that their model can generalize to datasets on which their model has not been trained on. To take one easy step in this direction, it would be good if the authors could train their model on CLEVR-Augmented and test how it can disentangle objects on the CLEVR-MV and vice versa. - The authors do not show any random scene generation results which makes it hard to judge whether or not the model has truly learned the underlying data distribution

Correctness: Overall the authors seem to make realistic claims about their model given the results they show

Clarity: The paper does not have a good clarity. The first two sections are pretty bad, specially the introduction.

Relation to Prior Work: The authors have been able to cover prior works and motivate their work but the related works section needs more work. The authors should also cite some more works which I pointed to.

Reproducibility: Yes

Additional Feedback: [Additional Comments] Abstract: - The abstract starts by focusing on the technical shortcomings of prior works and ends with summary of the experiments. The abstract should also include one or two sentences that describe the high-level motivation behind the work at the beginning. Before talking about the experiments the authors should also briefly describe 1) what the core component of their approach is (e.g. we replace the code layer of VAE with a more structured representation in this way or that way) 2) what’s the benefit of their approach compared to other works and 3) how it helps achieve their goal (e.g. it helps us do task X and task Y) Introduction: - The introduction is not written well and has lots of repeated materials specially in the 3rd and 4th paragraphs. [Missing References] - SOSV: Wu, Jiajun, et al. "Learning a probabilistic latent space of object shapes via 3d generative-adversarial modeling." Advances in neural information processing systems. 2016. - SOSV & SOMV: Rezende, Danilo Jimenez, et al. "Unsupervised learning of 3d structure from images." Advances in neural information processing systems. 2016. - SOSV & SOMV: Soltani, Amir Arsalan, et al. "Synthesizing 3d shapes via modeling multi-view depth maps and silhouettes with deep generative networks." Proceedings of the IEEE conference on computer vision and pattern recognition. 2017. - MOSV: Anciukevicius, Titas, Christoph H. Lampert, and Paul Henderson. "Object-Centric Image Generation with Factored Depths, Locations, and Appearances." arXiv preprint arXiv:2004.00642 (2020). [Rebuttal (prioritized)] 1- I would like to see results for scene factorization, novel view predictions and disentanglement analysis when the model has access to only 1, 2 and 3 observations during test time. It would be great if the authors can provide a video for this. It would be more ideal if they can also show the results obtained from GQN and IODINE. 2- Since the authors are training a generative model, one would naturally expect to see random samples from that generative model. Why the authors do not show samples in the paper or supplementary materials? Can the trained model be used to generate random scenes? If so, I would like to to see some randomly generated scene from the models trained on the CLEVR-MV and GQN-Jaco datasets. 3- It is not clear to me how many views the model use sees during training. Can the authors elaborate on this? For instance, in the supplementary materials the authors say they use 11 observations from the 20 available ones. Does that mean during each gradient step is always performed on these 11 randomly-chosen observations? If the answer is yes, could the authors say what this number is for other datasets? If the answer is yes, I would like to see how the model performs if it is trained using less number of observations. 4- A video that shows segmentations when moving the camera on the CLEVR-MV and GQN-Jaco datasets.


Review 2

Summary and Contributions: Implicit scene representations are a rising field with currently limited applicability. One of the major drawbacks is that it is usually just one object, or when multiple objects are present the viewpoint is fixed. This paper proposes a multi-view and multi object implicit scene representation framework. In the experiment they show generalization for novel viewpoints (however no generalization for novel objects or scenes). This paper got overall pretty good reviews, and I personally think that most of our concerns are either adressed in the rebuttal and will be fixed in the camera ready version. Reviewer 4 was the only one giving a slightly negative rating, but with the promise to raise the score if their concerns are adressed - let's see what you think :) I would argue to accept the paper, especially since two of us voted for a clear accept and all of us agree (including reviewer 4 with a marginally below acceptance rating) that the topic is important, the experiments are good and that the results are impressive.

Strengths: The paper approaches a currently unsolved hard problem and proposes a nice approach. Especially the disentanglement results are impressive. It shows clear benefits over MONet and IODINE. The choice of experiments seems legit

Weaknesses: It looks like there are some issues that might arise from writing a paper last minute. lines 28-34 are nearly identical to lines 35-43 and there are some obvious typos. My main concern would be that it solves only the novel viewpoint challenge and not generalization to new scenes. The disentanglement however is impressive.

Correctness: I honestly don't understand all the details, but what I understand seems ocorrect

Clarity: The paper is well written, has some minor issues that were stated before under weakness (can easily be fixed for camera-ready version)

Relation to Prior Work: A lot of effort was put into carefully asses prior work and this seems clear and correct to me.

Reproducibility: Yes

Additional Feedback: typos: intelligence.In )they I usually cite Yuille for vision as inverse graphics (since the idea is not that novel as indicated by the two citations you choose) @article{yuille2006vision, title={Vision as Bayesian inference: analysis by synthesis?}, author={Yuille, Alan and Kersten, Daniel}, journal={Trends in cognitive sciences}, year={2006} }


Review 3

Summary and Contributions: This paper proposes a method called MulMON for learning multi-object scene representations from multiple views, based on iterative amortized inference. MulMON can be seen as combination of the recent IODINE framework and GQNs.

Strengths: The problem multi-object representation learning is important, and the ability to use and generate different views of a scene is very relevant for the community. The proposed framework is well motivated, theoretically sound and the method is an elegant extension of IODINE. Evaluation is thorough and the results are impressive, especially on the GQN-Jacko dataset.

Weaknesses: I don't have any major complaints about this work. My main issues revolve around clarity.

Correctness: Object segmentation scores are slightly misleading. The stark difference in mIOU between IODINE and MulMON is mostly due to different handling of the background, which is arguably not that important. When ignoring background IODINE obtains close to perfect scores on vanilla CLEVR. I think the authors should include a score that ignores background, or at the very least discuss this difference.

Clarity: The paper is understandable, but contains several minor mistakes and some explanations are confusing. For example: - A few missing spaces (eg. lines 12 and 22). - Double use of iteration for both refinement steps and multi-views is confusing (eg. lines 160 and 163). - line 85 "... as discussed in Section 1": cannot find a discussion of this point in Section 1. - The derivation in line 116ff is confusing: In what sense is the set z = {z_k} grouped into T groups based on the views? Isn't each of the k objects present in each of the T views? I think the division into t observations is just an choice for the inference model, so in essence the assumption is that z = z^T, which implicitly depends on z^1, ... z^T-1. - Figure 2b) I don't think there should be a minus sign in f_\Phi (x^t, - v^t).

Relation to Prior Work: Recently there has been increased interest in object representation learning, with many new methods being suggested. May be worth mentioning some of them, especially SPACE [1]. [1] Lin, Zhixuan, et al. "Space: Unsupervised object-oriented scene representation via spatial attention and decomposition." arXiv preprint arXiv:2001.02407 (2020).

Reproducibility: Yes

Additional Feedback: Best discussion of the broader impact I have read so far.


Review 4

Summary and Contributions: This paper proposes a framework for learning the scene representations from the multi-object multi-view observations. The model is a natural extension from previous work and achieves good performances during the experiments.

Strengths: The paper is well-written, including the motivations and clear comparisons with previous work. The experiments are clearly conducted and the method outperforms previous methods. The multi-object multi-view problem in learning the scene representation is a missing piece and an important topic as well, which is a key step for moving toward to more complex scenes. This paper could be a good work in this direction.

Weaknesses: One concern I have is that compared with the novelty in addressing this new problem, the contribution in the learning method seems not enough. Most of the method are based on the previous work (IDINE). I am not sure whether the overall contribution is sound enough for this topic and for the neurips conference. Since the method can learn a composition of objects, it remains unclear how this learned composition can be applied or generalized to new scenes. These capabilities are critical in the task of learning from multi-objects. I hope to see more analysis on the compositions of objects besides the disentanglement analysis. I am willing to raise my score if these two concerns can be addressed. I read the review and rebuttal, they addressed most of my concerns. I raised the score accordingly.

Correctness: Yes

Clarity: Yes

Relation to Prior Work: Yes

Reproducibility: Yes

Additional Feedback:

[Author Response · NeurIPS 2020]

*Overview.* We would like to thank all reviewers for their thorough reviews and helpful feedback/suggestions.

*Generalization Experiments.* Three of four reviewers asked about generalization to novel objects and scenes. To
address this, we trained MulMON, IODINE and GQN on CLE-Aug. Then, we compared their performance on CLE-MV
and 2 new datasets—Black-Aug and UnseenShape. Black-Aug contains the CLE-Aug objects but only in single, unseen
colour (black). This tests the models' ability perform segmentation without colour differences/cues. UnseenShape
contains only novel objects that are not in the CLE-Aug dataset—cups, cars, spheres, and diamonds. This directly
tests generalization capabilities. Both datasets contain 30 scenes (more for final paper), each with 10 views. Table 1
shows that 1) all models generalize well to novel scenes, 2) MulMON still performs best for all tasks but observation
prediction—where GQN does slightly better due to its more direct prediction procedure (features $\rightarrow$ layout vs. features
$\rightarrow$ objects $\rightarrow$ layout), 3) MulMON can indeed understand the composition of novel objects in novel scenes—impressive
novel-view predictions (observations and segmentations) and disentanglement.

**Black-Aug**      **UnseenShape**

| Tasks | Models | CLE-Aug (train) | CLE-MV | Black-Aug | UnseenShape |
|---|---|---|---|---|---|
| Seg. (mIoU) | IODINE | $0.51 \pm 0.001^{\star}$ | $0.61 \pm 0.002$ | $0.50 \pm 0.006$ | $0.51 \pm 0.004$ |
| | **MulMON** | **$0.71 \pm 0.000$** | **$0.71 \pm 0.004$** | **$0.67 \pm 0.002$** | **$0.64 \pm 0.004$** |
| Pred.Obs (RMSE) | GQN | $0.15 \pm 0.000$ | **$0.15 \pm 0.001$** | **$0.24 \pm 0.003$** | **$0.17 \pm 0.002$** |
| | **MulMON** | **$0.07 \pm 0.000$** | $0.16 \pm 0.002$ | $0.26 \pm 0.002$ | $0.21 \pm 0.006$ |
| Disent. (D,C,I) | IODINE | $0.54, 0.48, 0.21$ | $0.14, 0.12, 0.26$ | $0.2, 0.26, 0.27$ | $0.13, 0.12, 0.26$ |
| | **MulMON** | **$0.63, 0.54, 0.68$** | **$0.52, 0.48, 0.63$** | **$0.55, 0.55, 0.66$** | **$0.5, 0.47, 0.67$** |
| Pred.Seg (mIoU) | **MulMON** | **$0.69 \pm 0.001$** | **$0.71 \pm 0.004$** | **$0.68 \pm 0.005$** | **$0.60 \pm 0.005$** |

**Table 1**: Models' generalization performance. $^{\star}$Paper correction. **Figure 1**: Samples from novel-scene datasets.

*Writing.* We would like to thank the reviewers for feedback on writing style, including both structuring and typos,
as well as related work. In particular, we would like to thank R1 for detailed suggestions to improve the abstract,
introduction and related work sections. These have all been incorporated, and have greatly improved the paper's clarity.

**R4:** *'the contribution in the learning method seems not enough. Most of the method are based on the previous work*
*(IODINE)'.* IODINE's approximation of the MOSV posterior cannot maintain object correspondence or "matching"
across multiple views, and thus, it is not a feasible solution to the MOMV problem. To address this, we make two
contributions in the learning method itself: 1) we side-step the object matching problem by iterating over multiple views,
each time using the previous iteration's approximate posterior as the new prior—see eq. 4; 2) we introduce a training
procedure that forces the model to make use of these iterative updates, aggregating spatial information across multiple
views. Specifically, by randomly partitioning the views of a scene into two subsets, we can ask the model to predict
the views in one (novel viewpoint-queried generation) having observed the views in the other (scene learning)—see
eq. 5. This forces the model to use the iterative updates to aggregate spatial information across multiple views, as it
needs to form a complete 3D scene understanding in order to perform well. If instead we had asked the model to simply
reconstruct the observed view, as IODINE does, it would completely overwrite the prior on each iteration—as it would
not need to aggregate spatial information across multiple views in order to perform well.

**R1:** *'results for scene factorization, novel view predictions and disentanglement. . . only 1, 2 and 3 observations'.*
Figure 3 in Appendix E plots novel-viewpoint observation and segmentation results as a function of the number of
views $T$ (1-10). However, we appreciate R1's suggestion of adding baselines (IODINE, GQN) to these analyses, and
also analyzing both segmentation and disentanglement results as a function of the number of views $T$. We will definitely
include these in the final paper. *'Can the trained model be used to generate random scenes?'* Yes, we can generate
random scenes by composing independently-sampled objects. However, to focus on forming accurate, disentangled
representations of multi-object scenes, we must assume objects are i.i.d. and thus ignore inter-object correlations—e.g.
two objects can appear at the same location. Nonetheless, we will include random samples in the final paper. *'not clear*
*. . . how many views the model sees during training'.* Using a fixed number of observations could harm the model's
robustness at test time. Thus, the number of observations given to the model is sampled at every training step—from the
range $[1, 6]$ for the GQN-Jaco dataset, and from $[1, 5]$ for the CLEVR-based datasets. Then, the model is asked to predict
the remaining, unseen views. Note that GQN-Jaco dataset has a total of 11 views, while the CLEVR-based datasets
have 10. This sampling procedure is introduced as a *randomized partition of $T$ observations into two subsets $\mathcal{T}$ and*
$\mathcal{Q}$ in lines 191-192 in the paper, but we will revise the text to further clarify this. *'A video that shows segmentations*
*when moving the camera. . . '* As we cannot include external links here, we will create one for the final version.

**R3:** *'include a score that ignores background, or ... discuss this difference'.* We thank R3 for pointing this out, and
we agree that IODINE's poor mIoU is mostly due to its handling of the background. We will discuss this issue in the
paper, and add a score (e.g. Adjusted Rand Index) to our comparisons that ignores the background. However, good
models shouldn't split up such simple backgrounds—see Fig. 3 in our paper and Fig. 1 in IODINE paper. *'In what*
*sense is the set $\mathbf{z} = \{z_k\}$ grouped into T groups based on the views?'* R3 is correct in that this design is just to show
that we update $\mathbf{z}$ one piece (i.e. $\mathbf{z}^t$) at a time w.r.t. its corresponding $x^t$. We will add some discussions to clarify this.

[Meta-Review · NeurIPS 2020]

This paper presents a GQN-like extension of IODINE with a small innovation in the iterative inference procedure whereby the model carries over the results of last iterative inference as a prior for the next view. The experimental results are very encouraging. Reviewers raised several clarity issues with the main text.